# Unraveling Neural Cellular Automata for Lightweight Image Compression

## Abstract

Neural Cellular Automata (NCA) are computational models inspired by cellular growth, capable of learning complex behaviors through local interactions. While NCAs have been applied to various tasks like image restoration and synthesis, their potential for image compression remains largely unexplored. This paper aims to unravel the capabilities of NCAs for lightweight image compression by introducing a Grid Neural Cellular Automata (GNCA) training strategy. Unlike traditional methods that depend on large deep learning models, NCAs offer a low-cost compact and highly parallelizable alternative with intrinsic robustness to noise. Through experiments on the COCO 2017 dataset, we compare the compression performance of NCAs against JPEG, JPEG-2000 and WebP, using the metrics PSNR, SSIM, and MSE and Compression Rate. Our results demonstrate that NCAs achieve competitive compression rates and image quality reconstruction, highlighting their potential as a lightweight solution for efficient image compression. The code will be available upon acceptance.

## 1 Introduction

Cellular Automata (CA) (Von Neumann, 2017) are computational models inspired by the process of cellular growth and differentiation, where these cells evolve over discrete time steps according to predefined rules and neighbouring cell states. Originating from the pioneering work of John von Neumann and Stephen Wolfram, with Conway's Game of Life (Adamatzky, 2010), Cellular Automata have found applications across various fields, including physics, biology, computer science, and social sciences (Hoekstra et al., 2010), due to their ability to capture and simulate a wide range of phenomena with simple yet powerful computational principles.

Building on the foundational concepts of traditional Cellular Automata, Neural Cellular Automata (NCA) introduces an advanced approach by integrating Deep Neural Network architectures to define their transition functions. Unlike CAs, the updated rules are not predefined but learned through training. Applications of NCA (Mordvintsev et al., 2020; Palm et al., 2022) have demonstrated the ability to reconstruct an image from a pixel seed by learning local neighbour interactions in a self-organized manner. Although many promising applications have been proposed, ranging from content generation (Sudhakaran et al., 2021) to self-organized control (Variengien et al., 2021), NCAs have not been explored in the context of data representation for image compression.

State-of-the-art (SOTA) methods for image compression typically rely on large deep neural networks (Zhu et al., 2022; Sadeeq et al., 2021). While autoencoders have been successfully used for image compression (Liu et al., 2022), these models often come with significant storage demands due to their size, which is frequently overlooked when calculating the storage size for compressed representations. By contrast, NCAs present a lightweight alternative with several compelling properties, such as asynchronous operation with local interactions, high parallelization, and noise tolerance due to their self-organized reconstruction capabilities.

This work proposes a novel approach to image compression using Neural Cellular Automata (NCAs). The key advantages of this approach include its parallelization capability, robustness to data corruption, and model compactness. Unlike traditional neural network-based solutions where the compressed representation is a latent vector, the proposed solution based on NCAs use the model weights themselves as the compressed data, reconstructing the image from a single pixel seed. The central research question we address is: : *How efficient are Neural Cellular Automata for Image*

*Compression*? To answer this, we utilized the NCA model proposed by Mordvintsev et al. (2020) and compared its performance against classical compression methods such as JPEG (Wallace, 1991), JPEG-2000 (Marcellin et al., 2000) and WebP (Si & Shen, 2016). We assessed the quality of compressed images using the metrics Peak Signal-to-Noise Ratio (PSNR) (Shirani, 2008), Structural Similarity Index Measure (SSIM) (Ding et al., 2020), Mean Squared Error (MSE) Thompson (1968) and Compress Rate (CR). The experiments were conducted on images from the COCO 2017 dataset (Lin et al., 2014) .

The main contributions of this paper are:

- To the best of our knowledge, our work is the first to investigate the potential use of Neural Cellular Automata for image compression, identifying benefits and limitations compared to traditional methods;

- We propose a grid training strategy to compress high-resolution images using NCAs, demonstrating high parallelization performance;

- We investigate NCA parameters for image compression and highlight NCAs as a novel paradigm for compact image representation.

## 2 BACKGROUND

Neural net-based formulations of CAs can be traced back to the early work of Wulff & Hertz (1992). Recent formulations of NCAs have shown their ability to learn complex desired behaviour, such as semantic segmentation (Sandler et al., 2020), common reinforcement learning tasks (Variengien et al., 2021), 3D locomotion (Najarro et al., 2022), Atari game playing (Najarro et al., 2022), and image synthesis (Palm et al., 2022).

Works by Mordvintsev et al. (2020) and Palm et al. (2022) show the capability of NCAs to reconstruct an image from a seed through learned neighbour interaction. Menta et al. (2024) uses NCA for resource-efficient image restoration, showing its potential for noise tolerance. In the context of image compression, the early work of Paul et al. (1999) proposed using Cellular Automata transformations to compress images. However, CAs have not been investigated as an efficient compression method since then.

Recent advancements in image compression have been driven by deep learning techniques (Mishra et al., 2022). Various compression algorithms based on convolutional neural networks (CNN) (Li et al., 2021), recurrent neural networks (RNN) (Medsker & Jain, 1999), autoencoders (Liu et al., 2022), and generative adversarial networks (GAN) (Gui et al., 2021) have shown significant improvements over traditional codecs like JPEG and WebP, particularly in terms of perceptual quality and rate-distortion performance (Sadeeq et al., 2021).

Using autoencoders and deep neural networks for image compression has some disadvantages despite their potential for high compression ratios and improved perceptual quality. These drawbacks include the need for large amounts of training data, large models, and significant processing power and memory resources for training. The SOTA MLIC++ (Jiang & Wang, 2023) uses a model with 83.5 million parameters, and the model size is not considered when calculating the compressed storage size, although these parameters are necessary to reconstruct the image. Large architectures like these demand extensive computational resources to train, encode and decode images (Mishra et al., 2022), making them unsuitable for lightweight devices.

This work proposes the use of NCAs to provide a lightweight solution with high parallelization capabilities, learning to compress and reconstruct images as self-organizing structures. Unlike autoencoders, which generate a latent compressed representation, we propose that NCAs use the model weights as encrypted, compressed representation form. This approach enables the image to be reconstructed from a pixel seed using the trained model, positioning the model itself as the compressed representation. To the best of our knowledge, this is the first study to explore NCAs for image compression.

## 3 METHOD

### 3.1 PROBLEM DEFINITION

Formally, image compression can be defined as a function $\mathcal{C} : \mathcal{I} \to \mathcal{C}(\mathcal{I})$ that maps an original image $I$ to a compressed representation $C(I)$, where $C(I)$ contains fewer bits than the original image while preserving important perceptual or structural features.

The goal of image compression is to minimize the distortion introduced during the compression process while significantly reducing data size. Image compression algorithms aim to minimize some distortion metric $\mathcal{D}$ under a constraint on the size of $C(I)$. This can be formulated as an optimization problem $\min_{\mathcal{C}} \mathcal{D}(\mathcal{I}, \mathcal{C}(\mathcal{I}))$ subject to a size constrain on $\mathcal{C}(\mathcal{I})$, where $\mathcal{D}(\mathcal{I}, \mathcal{C}(\mathcal{I}))$ represents the distortion between the original and compressed images.

### 3.2 NEURAL CELLULAR AUTOMATA (NCA)

Cellular Automata (CA) are defined as a tuple M = (L, S, $\eta$, $\phi$), where $L$ is a d-dimensional lattice, $S$ is the set of possible states, $\eta : L \to 2^L$ is the neighbourhood function and $\phi : S^n \to S$ is the transition function. Let $s_i^t \in S$ indicate the state of cell $I$ at time step $t$ and $\Omega_i^t = s_i^t : j \in \eta(i)$ the set of neighbours of the cell I according to $\eta$. Then, the state update for cell $i$ at each iteration $t$ is computed as $s_i^{t+1} = \phi(s_i^t \cup \Omega_i^t)$.

Neural Cellular Automata (NCAs) extend the CA framework by replacing the pre-defined transition function $\phi$ with a learnable function represented by a neural network $\phi_\theta$ with parameters $\theta$, preserving the essential feature of the locality. NCA can be trained to reconstruct images starting from pixel seeds, which means to start the reconstruction from a blank image with a seed pixel and, through learned neighbour rules, reconstruct the entire image (Mordvintsev et al., 2020), as shown in Figure 1. The original NCA approach trains a single model for each image (Mordvintsev et al., 2020), which is a lightweight model by design. Although alternative methods have been proposed to train one model for multiple images, they often require larger models or external support (Hernandez et al., 2021), which we do not explore in this work.

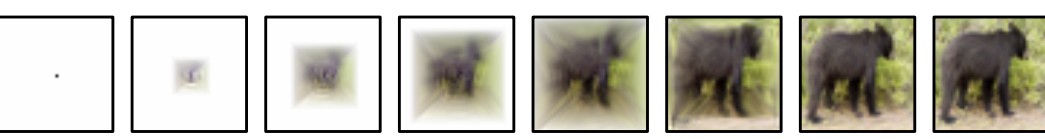

Figure 1: Reconstruction of an image using a trained NCA, starting from pixel seed. The figure shows reconstruction through iterations.

The NCA, as defined by Mordvintsev et al. (2020), starts from a constant zero-filled initial state called seed and evolves this state over time according to its update rule. In this model, each cell is composed of a $C$-depth state. The update rule of this NCA consists of two parts: Perception and Stochastic Update. At the perception stage, each cell gathers information from its surrounding neighbours, forming the perception vector $z_{ij} \in \mathbb{R}^{3C}$, defined as:

$$z_{ij} = \text{concat}(s_{ij}, K_x * s_{ij}, K_y * s_{ij}),\tag{1}$$

where $K_x$ and $K_y$ are predefined Sobel filters related to $x$ and $y$ axes, and $*$ stands for 2D-depthwise convolution operation using a $3 \times 3$ kernel size.

The perceived information is passed through a Multilayer Perceptron (MLP) (Almeida, 2020) neural network to compute the new state $s_{ij}^{t+1} = \text{MLP}(z_{ij}^t)$. The NCA is trained using backpropagation through time (BPTT) to compute gradients of the loss concerning the neural network weights. The reconstruction loss is defined by the mean squared error between the final state $S^t$ and target state $S^*$, as follows:

$$\mathcal{L} = \frac{1}{T} \sum_{t=1}^{T} \mathcal{L}^t,\tag{2}$$

$$\mathcal{L}^t = \frac{1}{HW} \sum_{i,j} (S_{ij}^t - S^*)^2, \tag{3}$$

where $H$ and $W$ are the height and width of a 2-dimensional lattice.

### 3.3 NCA for Image Compression

We propose using NCA as a compression model, as they can reconstruct an image from a pixel seed using a lightweight model. Different from Autoencoders, which extract a compressed latent representation of an image, in the proposed application of NCA, the compressed representation is the model itself since it can reconstruct the image from a pixel seed. As proposed by Mordvintsev et al. (2020), the NCA requires a different model for each input image to generate the compressed representation. Although some NCA variations enable one model to build many images (Menta et al., 2024), they require larger models and are not considered in this analysis. As the NCA proposed by Mordvintsev et al. (2020) is a low-resource model, the size of the required weights to reconstruct the image is lower than the input image size. Therefore, it is capable of generating a compressed and efficient representation.

When defining an NCA architecture for image compression, the objective is to define a model with the lowest number of parameters but the highest reconstruction quality. From a compression perspective, the compression rate is based on the size of the model weights compared to the size of the input image. The total number of parameters $n_p$ used in this model is defined as follows:

$$n_p = h_s \cdot (4 \cdot n_c + 1), \tag{4}$$

where $h_s$ is the hidden size (i.e. the number of neurons in the hidden layer) and $n_c$ is the number of channels or states for each cell.

Reconstruction quality depends mainly on the parameters $h_s$ and $n_c$, as will be shown in section 5.1. The main problem is that the number of steps necessary to reconstruct the image increases as the image size increases. Thus, backpropagation through time will be longer, implying more training time and a tendency for vanishing gradients. Also, larger images will require a larger number of parameters as the number of reconstruction patterns increases.

To address this problem, we propose the Grid Neural Cellular Automata (GNCA) training strategy for image compression. In this approach, we divide the input images into patches with size $P_w \times P_h$, where $P_w$ and $P_h$ are the width and height of each patch. For each patch, we train a different NCA to reconstruct that patch. The compressed representation is defined by $R = \sum^n \text{size}(\theta_n)$, where $\theta_n$ are the parameters of the NCA model trained at patch $n$. Figure 2 shows the encoding stage to obtain the compressed representation. The decoding stage uses the trained model to rebuild each patch through inference. The merged patches reconstruct the entire image.

The GNCA approach addresses the scaling issue by using smaller models for each patch, which can be trained and reconstructed in parallel. This allows for efficient compression of large images while maintaining high reconstruction quality. With this, no matter the size of the original input image, a set of NCAs is able to learn to compress the representation of each image patch and then reconstruct it. The process of reconstructing an image starts by providing an empty image with a pixel seed to the trained model, as shown in Figure 1.

## 4 Experimental Environment

### 4.1 Dataset

For our experiments, we used images from the COCO 2017 dataset (Lin et al., 2014) for the experiments. This dataset includes a diverse range of images of complex everyday scenes containing common objects in their natural context, with images of different sizes. For our experiments, we selected 30 random images from the dataset. We used a reduced number of images because each image requires a different model to train. We resized the original images to the sizes $40 \times 40, 80 \times 80$ and $120 \times 120$ pixels to our experiments.

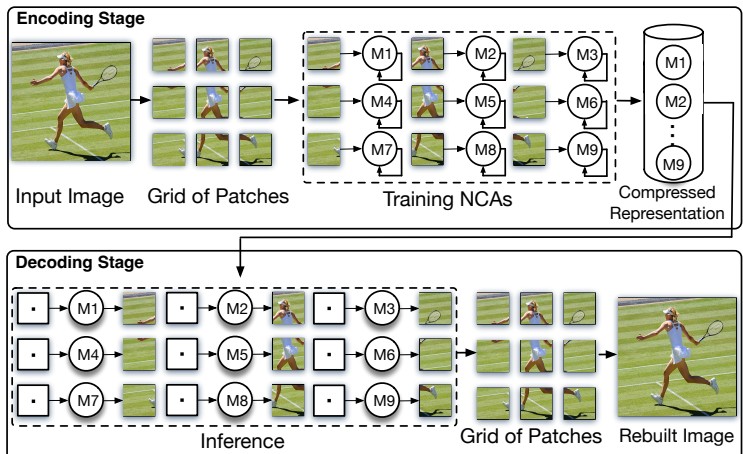

Figure 2: Encoding and decoding stages from Grid Neural Cellular Automata (GNCA). In the encoding stage, the image is divided into patches, which are trained by different NCA models, represented as *M*. The model weights are the resulting compressed representation. In the decoding stage, we use inference of each model to reconstruct the image.

## 4.2 METRICS

To asses the quality of the compressed image, we used the metrics Peak Signal-to-Noise Ratio (PSNR), Mean Squared Error (MSE) and Structural Similarity Index Measure (SSIM), which are metrics used in literature for evaluating the quality of compressed images.

MSE is a measure of distance between the reconstructed image and the original image. The smaller the MSE, the more similar are the images. MSE is defined as follows:

$$\text{MSE} = \frac{1}{MN} \sum_{i=1}^{M} \sum_{j=1}^{N} \left( I(i,j) - \hat{I}(i,j) \right)^2 , \tag{5}$$

where $I$ and $\hat{I}$ are the original and reconstructed image, with dimensions $M \times N$.

PSNR is calculated as the ratio of the maximum possible power of a signal to the power of corrupting noise that affects the fidelity of its representation. In the context of image compression, PSNR is often expressed in decibels (dB) and is calculated using the mean squared error (MSE) between the original and compressed images. Higher PSNR values indicate better image quality. The equation for Peak Signal-to-Noise Ratio (PSNR) is expressed as follows:

$$\text{PSNR} = 10 \cdot \log_{10} \left( \frac{I_{Max}^2}{\text{MSE}} \right) \tag{6}$$

where $I_{Max}$ is the maximum pixel value of the image.

SSIM assesses the visual impact of three characteristics of an image: luminance, contrast, and structure. The metric compares local patterns of pixel intensities that have been normalized for luminance and contrast. This makes SSIM particularly effective at modelling perceived changes in structural information, which are more important to human visual perception than changes in luminance or contrast alone. The equation for SSIM is given by :

$$\text{SSIM}(x,y) = \frac{(2\mu_x \mu_y + c_1)(2\sigma_{xy} + c_2)}{(\mu_x^2 + \mu_y^2 + c_1)(\sigma_x^2 + \sigma_y^2 + c_2)} \tag{7}$$

where $x$ and $y$ are the uncompressed and compressed images, respectively, $\mu_x$ and $\mu_y$ are sample means, $\sigma_x$ and $\sigma_y$ the standard deviations and $c_1$ and $c_2$ the stabilization coefficients. This metric

leverages a kernel to extract the structural information of images rather than focusing on single-pixel errors. This metric varies in the interval $[0, 1]$, with higher values being better.

We also evaluate the compress rate (CR) for each model. For the GNCA, we compare the original image size with the number of parameters of the model, considering a quantisation of one byte per parameter. Viewing the model weights as the compressed representation, the compression rate is defined by $CR = 1 - \frac{\text{Size}(\theta_I)}{\text{Size}(I)}$, where $I$ is the input image and $\theta_I$ the model parameters to reconstruct $\hat{I}$. When using quantised models, $\theta_I$ is defined by the number of parameters. According to Mordvintsev & Niklasson (2021), experimentation with quantisation of the NCA models to just one byte per parameter showed no evidence of degradation in quality.

### 4.3 IMPLEMENTATION

Our NCA model is based on the architecture proposed by Mordvintsev et al. (2020), and adapted for Pytorch. All experiments were conducted using an NVIDIA GPU 3090 (16GB) and Intel Core i5 processor. Each experiment consisted of training the model for 40,000 epochs, with a batch size of 8, a learning rate of 2e-3, and 224 iterations per epoch. For every image, we trained a separate model with learned parameters specific to that image. The patch size for all images was set to $40 \times 40$ pixels. The original NCA supports both synchronous and asynchronous training, through the parameter fire rate. For our experiments, we set the fire rate to 0 to train synchronously, as this configuration leads to faster convergence. However, asynchronous training resulted in similar performance but with increased training times.

We proposed three versions of GNCA, using low, medium and high amounts of parameters, to be competitive with the compressed image size of the compared methods. These versions vary the number of neurons in the hidden layer $h_s$ and the number of channels $n_c$, producing the total number of parameters, as defined in Eq. 4. The parameters and corresponding model sizes are shown in Table 1.

Table 1: GNCA configuration used in the experiments.

| Name | $n_c$ | $h_s$ | #Params |
|---|---|---|---|
| GNCA-small | 14 | 20 | 1140 |
| GNCA-medium | 16 | 32 | 2080 |
| GNCA-large | 16 | 40 | 2600 |

## 5 RESULTS

### 5.1 ANALYSIS OF NCA

We evaluated how the parameters of the NCA model influence its ability to reconstruct images and the total model size. Initially, we varied the hidden layer size and observed its effect on the PSNR for image reconstruction after model training. Figures 3(a) and 3(b) show how PSNR and the number of parameters increase as the size of the hidden layer increases, using a 40x40 pixel image from the COCO dataset. Figures 3(c) and 3(d) illustrate how the number of channels affects both the PSNR value and the number of parameters. By increasing the hidden layer and number of channels, we observed that the model learns to reconstruct the image more closely to the original one. However, increasing model size to improve image quality reduces the compression rate as the number of parameters increases. Thus, the tradeoff between image quality and compression rate should be tuned by adjusting the number of model parameters.

We also analyzed how the reconstruction quality of NCA scales with image size. Figure 4 shows the PSNR metric with different image sizes when training NCAs while keeping the same parameters. To maintain high PSNR as image size increases, NCA needs more parameters. However, using a single model for high-dimensional images may be infeasible due to high computational resource requirements for large models and the potential gradient vanishing problem as the number of back-propagation iterations increases. Using the GNCA strategy allows us to maintain the same model

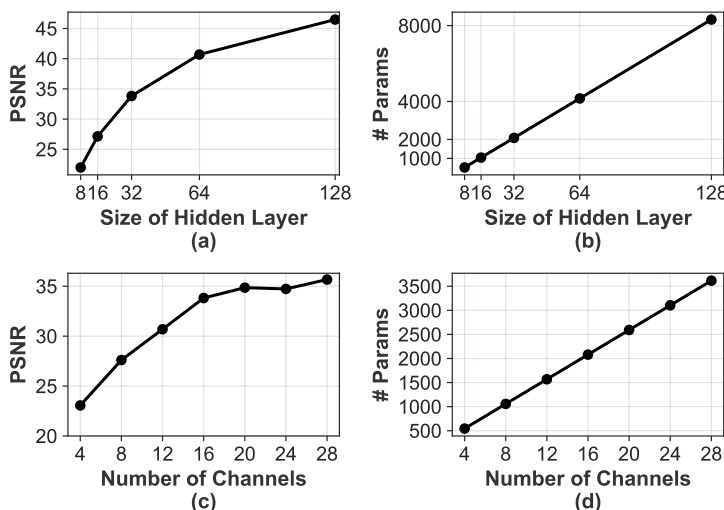

Figure 3: Analysis of (a) PSNR versus Size of Hidden Layer, (b) Number of Parameters versus Size of Hidden Layer, (c) PSNR versus Number of Channels and (d) Number of Parameters versus Number of Channels, when evaluating an NCA trained to reconstruct an image of size 40×40 from COCO dataset.

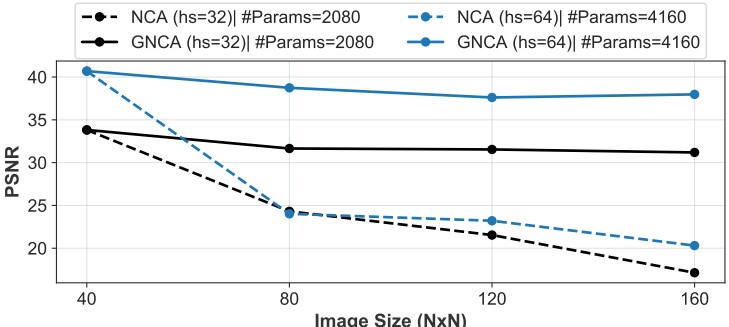

Figure 4: Analysis of PSRN metric versus the image size and the parameter *hidden size*. GNCA scales better as the image size increases compared with the standard NCA strategy.

size while preserving the PSNR value. The total compressed representation size is the sum of the parameters from the NCA models used for each patch.

We also evaluated the quality of reconstructed images using GNCA-small, GNCA-medium and GNCA-large. Figure 5 shows this comparison, for $120 \times 120$ images from COCO dataset. We can notice there is a tradeoff between model size and image quality, but it show the capability of the NCA to reconstruct the image from a single pixel, being close to the original image, using a lightweight model.

## 5.2 COMPARISSON WITH COMPRESSION METHODS

We evaluated the compression capabilities of NCA compared to JPEG, JPEG-2000 and WebP, using maximum quality settings for each method. Table 2 presents the average results of each method for compressing $40 \times 40$, $80 \times 80$ and $120 \times 120$ images. Results include average values of PSNR, MSE, SSIM and CR metrics.

The results demonstrate that GNCA achieves competitive results in terms of compressed image quality over differ image resolutions. GNCA-medium showed a good balance between image quality and compression rate, for $40 \times 40$ and $80 \times 80$ resolution. While JPEG and WebP achieved higher

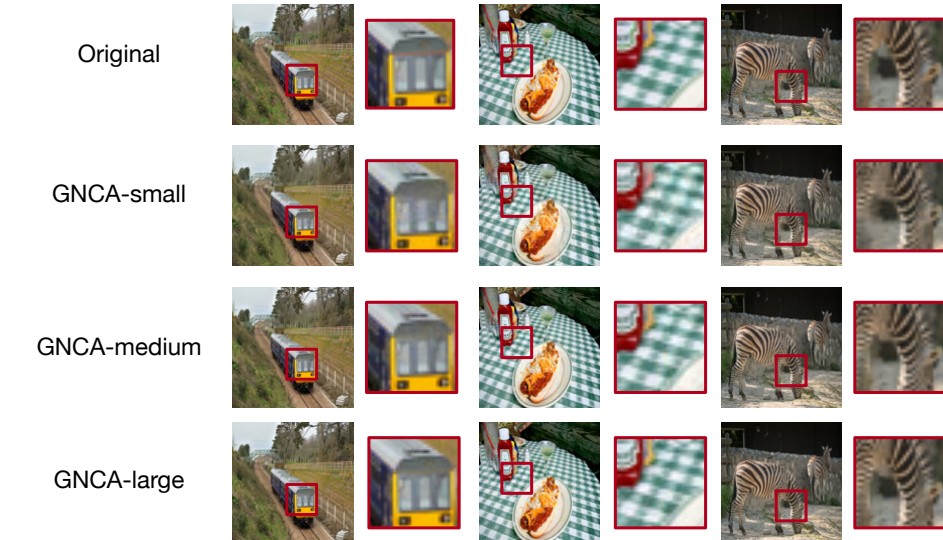

Figure 5: Visualization of reconstructed images using GNCA-small, GNCA-medium and GNCA-large, for $120 \times 120$ image size.

Table 2: Average results of image compression for COCO dataset using GNCA, JPEG, JPEG200 and WebP compression methods.

| Image Size | Method | PSRN↑ | MSE↓ | SSIM↑ | CR↑ |
|---|---|---|---|---|---|
| | JPEG | 32.72±2.91 | 43.58±31.06 | 0.97±0.01 | 62.84±13.72 |
| | JPEG-2000 | 33.28±3.09 | 38.73±29.50 | 0.96±0.02 | 66.16±10.60 |
| | WebP | 33.81±3.52 | 37.56±31.43 | **0.98±0.01** | 65.36±10.53 |
| $40 \times 40$ | GNCA-small | 28.06±3.14 | 129.65±92.61 | 0.90±0.05 | **75.98±7.40** |
| | GNCA-medium | 32.58±3.14 | 46.20±34.81 | 0.96±0.02 | 56.17±13.50 |
| | GNCA-large | **34.66±3.16** | **28.74±21.88** | 0.97±0.01 | 45.22±16.87 |
| | JPEG | 33.48±2.67 | 35.97±27.14 | 0.94±0.95 | 67.67±5.52 |
| | JPEG-2000 | 34.54±5.03 | 27.57±2.87 | **0.96±0.02** | 68.46±5.04 |
| | WebP | 35.42±2.55 | 22.40±11.81 | 0.97±0.12 | 67.60±4.07 |
| $80 \times 80$ | GNCA-small | 30.34±2.72 | 71.64±40.64 | 0.89±0.04 | **68.85±5.24** |
| | GNCA-medium | 34.75±2.57 | 25.38±12.86 | 0.95±0.03 | 43.17±9.56 |
| | GNCA-large | **36.06±2.37** | **18.24±8.60** | **0.96±0.03** | 30.81±12.79 |
| | JPEG | 33.81±2.74 | 34.22±29.26 | **0.98±0.01** | **69.55±3.87** |
| | JPEG-2000 | 36.23±2.66 | 18.30±10.63 | 0.97±0.01 | 65.95±4.02 |
| | WebP | 34.87±3.23 | 29.29±29.50 | **0.98±0.01** | 66.64±2.50 |
| $120 \times 120$ | GNCA-small | 30.79±2.51 | 63.62±36.81 | 0.94±0.03 | 66.52±4.37 |
| | GNCA-medium | 35.49±2.50 | 21.46±12.05 | **0.98±0.11** | 38.91±7.98 |
| | GNCA-large | **37.24±2.54** | **14.50±8.49** | **0.98±0.01** | 23.64±9.98 |

compression rates for the similar PSNR values, the GNCA models offer flexibility in enhancing image quality or compression rate (defined by the number of GNCA parameters) by tuning the hidden layer size, as shown in Figure 4.

In this study, we applied the Friedman test (Friedman, 1937) and the pairwise Mann-Whitney U test (Sheskin, 2003) to evaluate whether there are statistically significant differences among the performance of the different image compression methods evaluated. The Friedman test was initially conducted to detect any overall differences across the methods. The results indicated a statistically significant difference among the groups. To identify which specific methods differed from each other, we performed post-hoc pairwise comparisons using the Mann-Whitney U test. The results of these tests revealed significant differences between several pairs of methods, as shown in Table 3.

No significant differences were found between JPEG, JPEG-2000 and WEBP when compared to GNCA-medium int terms of PSNR metric. c.

Table 3: Comparison of Methods after using Pairwise Mann-Whitney U Test. The symbol ✓ means there is a statistically significant difference ($p-value < 0.05$) between the methods for the evaluated metrics.

| Image Size | $40 \times 40$ | | $80 \times 80$ | | $120 \times 120$ | |
|---|---|---|---|---|---|---|
| Comparison | PSNR | CR | PSNR | CR | PSNR | CR |
| GNCA-small vs JPEG | ✓ | ✓ | ✓ | X | ✓ | X |
| GNCA-small vs JPEG-2000 | ✓ | X | ✓ | X | ✓ | X |
| GNCA-small vs WebP | ✓ | X | ✓ | X | ✓ | X |
| GNCA-small vs GNCA-medium | ✓ | ✓ | ✓ | ✓ | ✓ | ✓ |
| GNCA-small vs GNCA-large | ✓ | ✓ | ✓ | ✓ | ✓ | ✓ |
| GNCA-medium vs JPEG | X | X | X | ✓ | X | ✓ |
| GNCA-medium vs JPEG-2000 | X | X | X | ✓ | X | ✓ |
| GNCA-medium vs WebP | X | X | X | ✓ | X | ✓ |
| GNCA-medium vs GNCA-large | ✓ | X | X | X | X | ✓ |
| GNCA-large vs JPEG | X | X | X | ✓ | ✓ | ✓ |
| GNCA-large vs JPEG-2000 | X | ✓ | X | ✓ | X | ✓ |
| GNCA-large vs WebP | X | ✓ | X | ✓ | X | ✓ |

In terms of encoding and decoding time, NCA requires model training to learn image compression for each image, making it suitable only for offline compression. The training time, using the setup described in section 4.3, is approximately 3.2 hours per patch, which can be trained in parallel. The inference time for reconstruction is 0.05 seconds.

## 6 DISCUSSION

Results demonstrate that NCAs can be used for data compression by treating the trained model weights as the compressed representation. The architecture of the model plays a crucial role in determining the compression efficiency, especially in balancing the tradeoff between the number of parameters and the quality of the reconstructed image. Our findings indicate that this approach is more efficient for low-resolution images, but the compression rate tends to decrease as image size increases, especially when compared to JPEG, JPEG-2000, and WebP. Despite this, the GNCA method shows competitive performance against these classical methods, particularly in scenarios that benefit from high parallelization.

Compared to SOTA neural network compression methods, the solution based on NCA proposes a much lighter approach. Our largest model (GNCA-large) uses only 2,600 parameters, compared to state-of-the-art deep learning-based compression methods, such as MLIC++ (Jiang & Wang, 2023), which requires 83 million parameters. This lightweight characteristic makes GNCA an attractive option for devices with limited computational resources, as it supports highly parallelized and asynchronous operations. However, due to the training time required for each image, NCAs are better suited for offline compression rather than real-time applications.

Some new approaches have been proposed to train a single model for many images (Hernandez et al., 2021), but they require larger models. Otimizing NCA architecture in future work can lead to more efficient compression. In terms of reconstruction quality, NCAs can generate images with higher quality than JPEG and WebP, given larger models. Below, we discuss the benefits and drawbacks of using NCA for image compression.

### 6.1 ADVANTAGES

The main advantages of using NCAs for compression are:

*Noise Robustness*: NCA models are naturally robust to noise and are often used in denoising applications. This analysis has been done by the work of Menta et al. (2024), where NCAs successfully

reconstructed images even when parts of the data were corrupted. The ability to reconstruct images from minimal information, such as a single seed pixel, further enhances this robustness

*Parallelization*: NCAs can easily be parallelized due to their localized operations. Each cell interacts only with its immediate neighbors, allowing for asynchronous updates across different parts of the image. This makes NCAs particularly well-suited for large-scale or distributed computing environments where parallel processing is essential.

*Low Computational Resources*: Compared to deep learning-based methods, NCAs require fewer parameters, making them highly efficient in terms of memory and computational power. This efficiency enables NCAs to be deployed on devices with limited resources, such as mobile or embedded systems, which are often unable to support large and complex neural networks.

*Predictable Compression Size*: Another advantage of NCAs is the ability to predict the size of the compressed representation before the training or encoding process begins. Since the compressed size corresponds directly to the number of model parameters, it is possible to determine the storage requirements beforehand, offering a level of control in memory-constrained environments.

Furthermore, the compact representation through model weights can serve as an encrypted compression.

### 6.2 LIMITATIONS

Despite the advantages, there are some limitations to the current NCA-based approach:

*Training Time*: NCAs require a significant amount of training time for each individual image, which limits their applicability in real-time compression scenarios. This makes the current approach more suitable for offline compression rather than dynamic, real-time environments.

*Model-per-Image Limitation*: In this work, we used NCA based on Mordvintsev et al. (2020), which trains one model per image. We chose this method because it provided higher-quality reconstruction with a low number of parameters. However, strategies to train one model to many images (Hernandez et al., 2021) can be explored in future works.

## 7 CONCLUSION

This work investigates the potential of Neural Cellular Automata for image compression. NCAs present compelling properties compared to SOTA approaches, such as low computational cost, asynchronous operation with local interactions, high parallelization, and robustness to noise. Our experiments compared NCA-based compression to traditional methods such as JPEG, JPEG-2000, and WebP, showing that NCAs can provide competitive image quality with far fewer parameters. We also introduce a grid strategy called GNCA to train NCAs in higher-resolution images.

Experiments show that NCA can store a compressed image representation with a high reconstruction quality. In this approach, the model weights store the compressed representation, whose size is smaller than the input image. NCA demonstrates competitive results compared to JPEG, JPEG-2000 and WebP, and it is capable of compressing and reconstructing images while maintaining a high level of similarity with the original image.

The key benefits of the proposed method include its lightweight model architecture, high parallelization capability, and robustness to noise. These properties make NCAs well-suited for use in devices with limited computational resources or environments that require asynchronous processing.

Although the current approach has some limitations—such as the need for long training times and separate models for each image—there is potential for optimization and improvement. Future work may focus on reducing the model size for higher-resolution images and developing methods to train NCAs for multiple images without significantly increasing model complexity. With these advancements, NCAs could emerge as a viable alternative to current state-of-the-art compression techniques, offering both efficiency and flexibility.

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
