# OpenReview forum: "Unraveling Neural Cellular Automata for Lightweight Image Compression"
_ICLR.cc/2025/Conference — Submitted to ICLR 2025_

### Official Review · Reviewer_hDrj · 2024-10-30

**Soundness:** 2
**Presentation:** 2
**Contribution:** 1
**Rating:** 3
**Confidence:** 4

**Summary:**

This paper tackles the problem of image compression through the usage of Neural Cellular Automata (NCA).

NCA is a recent computational paradigm that builds upon the much older model of Cellular Automata (CA), which consists of a connected lattice of stateful cells whose states are recurrently updated through the repeated localized application of an update rule. The update rule consists of two stages: an interaction stage, where for each cell, information is gathered from its neighbouring cells; and an update stage, where this information is processed to produce a cell update. While CA use a handcrafted update rule (a popular one being Conway's Game of Life), NCA use a learned update rule in the form of a simple neural net with convolutions in the interaction stage and an MLP in the update stage. The weights of the update rule are optimized via some downstream task, where the cell grid is evaluated against some target state after a certain number of cell updates. Depending on the way NCA are trained, they can end up being fairly robust lightweight models, able to correct themselves in the presence of adversarial attacks and adapt to out-of-distribution situations. Both CA and NCA have shown how repeated applications of lightweight/simple rules (whether learned or handcrafted) can result in fairly complex emergent behaviour.

For this paper, the authors propose to exploit key characteristics of NCA for image compression: adversarial robustness, high parallelization, asynchronous computability, high complexity through simplicity (i.e., model compactness).

The main proposed contributions of this paper are as follows:
1. The first to investigate the application of NCA for image compression.
2. A novel training strategy to address scaling issues related to compressing high resolution images.
3. An ablation on NCA parameters to highlight compression characteristics.

The authors compare their NCA (named Grid NCA, or GNCA) against common compression algorithms: JPEG, JPEG-2000, and WebP, on the COCO 2017 dataset on PSNR, SSIM, MSE, and Compress Rate (CR) metrics.

**Strengths:**

* Originality:
    * NCA research is still a relatively niche area within deep learning and so far no meaningful analysis has been made on their image compression capabilities. As far as I'm aware, the authors take the first step in investigating the application of NCA for image compression.
    * The main NCA advancement proposed: Grid-based NCA training, is an NCA training approach that has not been proposed in previous literature and shows promise in its usefulness in scaling up the training of NCA (specifically NCA where the central cell is the only seeded cell). It's a clever solution in dealing with training scalability that takes advantage of the parallel nature of NCA.
* Quality:
    * The authors make an honest attempt in highlighting the pros and cons of their NCA-based compression model, as listed in Section 6.2.
* Clarity:
    * For the most part, the submission is clear in describing the task at hand, the motivation (e.g., why use an NCA?), and related works. I'm pleased to see the authors cover related works on CA for image compression and NCA for image reconstruction. I'm also pleased to see them point out muNCA from Mordvintsev et al.

**Weaknesses:**

* Quality:
    * A max image size of 120x120 is quite small for the purposes of an image compression analysis. It would be nice to see larger images (e.g., HD or higher), as that's typically when image compression becomes useful in real-world applications.
    * I would suggest the following experiments to improve on the technical soundness of the paper (i.e., claims made on the efficiency of GNCA for image compression):
        * An examination of compression performance on larger image sizes, particularly HD resolutions (720p, 1080p, etc.)
        * An ablation over different patch sizes to analyze its effect on compression performance and training convergence.
        * Comparisons with current SOTA DL-based methods.
        * A comparison with an NCA where all cells are alive at all times (i.e., all cells are simultaneously seeded and can communicate immediately).
* Clarity:
    * The text does not provide a clear description of the composition of the NCA update rule. The authors use an instantiation of Mordvintsev's approach but without knowing specifics, it's difficult to verify equation 4. In this case, a figure showing the model architecture (input, output, layers used, dimensionality, etc.) would be helpful.
    * The training description (i.e., Section 4.3) is lacking useful information and is somewhat unclear in some areas. For example, was the pool-based training technique introduced by Mordvintsev et al. used here? Considering that you're not really training an NCA on a dataset but rather optimizing it on a single image, what's the meaning of an "epoch" and how does it relate to iterations (L285 and L286)? When is backprop through time (BPTT) performed? At the end of an epoch?
    * For Figure 3, it would be nice to see these results averaged over the COCO dataset instead of for a single image, as there could be large deviations in results depending on the image used.
    * A visual comparison with JPEG, JPEG-2000, and WebP would be helpful in Figure 5, as so far no qualitative comparison with those image formats have been provided.
    * It would be nice to see a comparison of compression and decompression speed across the various approaches.
        * The authors do provide a compression/decompression speed for GNCA in L455-457 but it would be helpful to compare this with competitors in some table.
* Significance:
    * Although the area of investigation is novel (analysis of NCA in the image compression setting), there is a lack of breadth and depth in the analysis and the results presented leave a lot to be desired, as it seems GNCA is not competitive at all with the competing handcrafted approaches and it's unclear how they stack up against other DL-based approaches.
    * I may be misinterpreting the results, but it seems that overall, JPEG, JPEG-2000, and WebP produce higher quality images at roughly the same compression rate as GNCA (L377-423). Considering that these image formats do not have to be trained, are incredibly quick to compress and decompress, and can have their compression rate flexibly changed (which L423-425 seem to imply is something GNCA can do but the others can't, which is not the case), it's difficult to see a motivation to use GNCA for image compression. So far, the experimental results do not motivate further investigation into using NCA for image compression.
    * Section 6.1 lists the advantages of GNCA and I struggle to see the validity behind some of the claimed advantages.
        * The first is "noise robustness", and this is something that is listed as an advantage but is never explored in the paper.
        * The second is "parallelization", which is not unique to NCA as all convolutional networks are parallel as well, including the competing handcrafted compression approaches. That being said, I do see an advantage with the "asynchronous" aspect of NCA, but unfortunately this is not a property that's well-investigated in this paper, so there's no motivation of its potential advantage.
        * The fourth is "predictable compression size". I don't see how this is a unique advantage of NCA compared to other approaches. It's all deterministic and so the compression size can be reliably estimated for the competing handcrafted approaches and for other DL-based approaches.

**Questions:**

* I'm curious why the "grow from a single seed" approach was used as opposed to having all cells seeded. In the former case, I can see why this led to implementing the grid-based training scheme, as it takes time for cells away from the center to come alive and begin to change state. However, in the latter case (which is what Mordvintsev et al. used in Self-Organizing Textures), that limitation is lifted and all cells are immediately alive and able to change state. In this latter case, one would presumably not need the grid-based training approach. My questions are:
    * Why choose the former approach (central cell starts off alive) over the latter (all cells are alive all the time)?
    * Have you tried the latter approach? If so, did you notice any benefit with the grid-based approach?
* I've mentioned some of these suggested experiments in the weaknesses section, but the ones in particular that I believe are absolutely necessary are:
    * An ablation over different patch sizes to analyze its effect on compression performance and training convergence.
    * Comparisons with current SOTA DL-based methods.
* Suggestions on improving the clarity of the paper:
    * Adding more information on the training section of the paper (4.3). Specifically, clarifying what an epoch and iteration represents considering that training is on a single image. Clarifying whether the "pool-based" training technique used in all of Mordvintsev et al's NCA approaches is used here. Clarifying when BPTT is performed in relation to iterations and epochs.
    * An architectural overview diagram that shows the input, output, layers used, and tensor shapes/dimensions. This will help in verifying equation 4.
    * Including qualitative results of JPEG, JPEG-2000, and WebP in Figure 5.
* Other suggestions:
    * An examination of compression performance on larger image sizes.
    * A comparison of compression and decompression speed across the various approaches presented in the paper.
    * For Figure 3, showing results averaged over the COCO dataset (or a significant subset) instead of for a single image.

---

### Official Review · Reviewer_DwoT · 2024-11-03

**Soundness:** 2
**Presentation:** 2
**Contribution:** 2
**Rating:** 5
**Confidence:** 4

**Summary:**

This paper proposes the use of Neural Cellular Automata (NCA) for image compression. The idea is to represent a given image as the result of a sequence of local updates parameterized by a neural network, whose parameters are learned by minimizing the distortion of the output w.r.t. a given image.  Compression is achieved by representing the image using a lightweight neural network, whose parameters can be further quantized. To achieve good reconstruction on larger images, the paper also proposes to apply NCA on patches, which is termed Grid Neural Cellular Automata (GNCA). Experiments are performed on COCO images of up to 120x120 using NCA with varying neural network sizes, and demonstrate that NCA with enough capacity can achieve reconstruction quality on par with conventional codecs (e.g., JPEG, WebP), although not necessarily better compression performance in terms of rate-distortion.

**Strengths:**

Originality:

The idea of using Neural Cellular Automata (NCA) for image compression is novel, to my best knowledge. By representing data as (approximately) an attractor of the reconstruction loss (reminiscent of a Hopfield network), the approach has some interesting characteristics, in particular "noise robustness" which might be useful for certain applications.  However besides the idea/framing, there seems to be little novelty in ML methodology, which is borrowed from Mordvintsev et al. (2020) (https://distill.pub/2020/growing-ca/).

Quality:

The quality is somewhat lacking; the experiments could have been more informative (see weakness below).


Clarity:

Writing is mostly clear and easy to follow, but some details are missing in the experiments (also see weakness below). I find some of the paper's claims not very well supported, again see below.

Significance:

Significance is somewhat low / medium. The paper mostly feels like a reproduction of the NCA work of [Mordvintsev et al. (2020)]((https://distill.pub/2020/growing-ca/)) and trying it on neural networks of a few different sizes, with the main contribution showing that NCA with bigger neural nets indeed yield better reconstruction quality (which can match JPEG, WebP, etc.). No effort is made to reduce the size of NCA parameters other than using a smaller network and/or 8-bit quantization.

**Weaknesses:**

There's a few issues that relate to the experiments:
1. Are the rate-distortion results reported using 8-bit quantized models?  It's unclear whether this is the case based on Lines 276 - 279.

2. It's unclear what / how many images are actually used in the evaluation. E.g., the caption of Table 2 refers to "Average results of image compression for COCO dataset". Average over how many images?

3. The paper argues that the NCA approach is more computationally lightweight than autoencoder-based approach, but I don't find this convincing without a comparison of decoding complexity, e.g., in terms FLOPs per pixel, which is standard in efficient neural compression research (see e.g., [Johnston et al.](https://arxiv.org/abs/1912.08771), [Yang et al.](https://arxiv.org/abs/2304.06244)). The paper's argument for NCA being more lightweight is mostly based on the size of the decoding program (e.g., "the solution based on NCA proposes a much lighter approach... uses only 2,600 parameters, compared to state-of-the-art deep learning-based compression method such as MLIC++ (Jiang & Wang, 2023), which requires 83 million parameter"), which is not very convincing given that the codec's storage size is typically amortized across many, many encoding/decoding operations and becomes negligible after a while.

4. Rather than Table 2, the results would be much more clearly presented using rate-distortion curves (e.g., PSNR v.s. bits per pixel), which is the standard in lossy compression literature.

5. Given the method is highly related to the NIR approach (e.g., COIN: COmpression with Implicit Neural representations, https://arxiv.org/abs/2103.03123), I think the latter can be reasonably considered a baseline. It seems to me the INR approach, which uses a feedforward network to compute the reconstruction, may be more efficient than NCA, which uses a recurrent (convolutional) network instead, especially in terms of training/encoding speed (no backprop across time needed).

More minor complaints:

6. I don't really agree with the argument that the approach offers "Predictable Compression Size" on line 497.
All the other approaches also have "Predictable Compression Size" in terms of the dimension of transform coefficients, if no entropy coding is applied. So this "advantage" isn't really a strength of the method itself.

7. I get the idea behind Eq (1) but as is written it doesn't really make sense, in particular the expression "K_x * s_{ij}" (convolving a filter with a pixel?). I think what is meant is the {ij}th coordinate of the convolution "K_x * s" between the filter $K_x$ and the entire image $s$, which itself is a function of the pixels in the neighborhood of s_{ij}.

8. Minor typographical issues on lines 123 through 126, e.g., different fonts for the cell index "I", "i".

**Questions:**

Some additional questions besides the ones above:

- Could the authors comment more on what "Noise Robustness" would mean in the context of compression? It's a little hard to imagine a situation in which the reconstruction pixels would be "corrupted" and need to be periodically "fixed".  In my view this "noise robustness"/fixed-point property seems to be the main (and perhaps only) selling point of an RNN-based approach compared to the feedfoward-network-based approach of autoencoders/INRs (as in the latter approaches have no recurrent/feedback connections in the decoding computation graph).

- This is more of a suggestion, but for the purpose of compression I don't think there's a strong reason to start the initial configuration of the NCA at a single fixed pixel, like in Mordvintsev et al. (2020). Other initial configurations may be more preferable, and could even be learned and transmitted like a latent code.

---

### Official Review · Reviewer_Htwm · 2024-11-03

**Soundness:** 2
**Presentation:** 1
**Contribution:** 2
**Rating:** 3
**Confidence:** 3

**Summary:**

This paper proposes a novel approach to image compression using Neural Cellular Automata (NCAs). To compress high-resolution images, a grid training strategy is proposed named Grid Neural Cellular Automata (GNCA). Experiments identifies benefits and limitations of GNCA compared to traditional methods.

**Strengths:**

1. Using Neural Cellular Automata for image compression is a novel idea in the field of image compression. And it is also an interesting idea.
2. To compress high-resolution images, GNCA is proposed which is a reasonable strategy. Experiments demonstrate its effectiveness.

**Weaknesses:**

1. The technique contribution of this paper is too limited. The GNCA training strategy is a trivial method. It just divides the input images into patches and trains different NCAs to reconstruct each patch. There is no information exchange between different pathes. Splitting images into patches is a common idea for the transformer model.
2. NCA suffers from inefficient training. Each image requires a different model to train. And each patch costs approximatedly 3.2 hours training time. Traditional methods do not need training and recent deep learning based methods train a model for all images. Compared with other methods, NCA is too inefficient to use in the application.
3. The high-resolution images used in this paper are 120x120 images. However, 120x120 cannot be considered as high-resolution in nowaday deep learning research.
4. Writing should be improved. In page 5, metrics are introduced too detailed. And please use RD-curve to compare performance of different methods.

**Questions:**

None

---

### Official Review · Reviewer_g4B5 · 2024-11-03

**Soundness:** 2
**Presentation:** 2
**Contribution:** 2
**Rating:** 3
**Confidence:** 2

**Summary:**

This paper explores the use of Neural Cellular Automata (NCA) for lightweight image compression by introducing proposing a Grid Neural Cellular Automata training strategy. The authors compare the compression performance of NCAs against JPEG, JPEG-2000, and WebP on the COCO 2017 dataset using diverse metrics. The experimental results demonstrate that NCAs achieve competitive compression rates and image quality reconstruction.

**Strengths:**

1. The paper introduces a new application of NCAs for lightweight image compression.
2. The GNCA strategy allows for efficient compression of large images by training smaller models for patches in parallel.
3. The paper presents a comprehensive experimental evaluation of GNCA and analyzes the impact of model parameters on compression performance.

**Weaknesses:**

1. While the paper compares NCAs with traditional compression methods, a direct comparison with state-of-the-art deep learning-based compression methods would provide a more comprehensive evaluation of NCA’s performance.
2. The computational cost of GNCA is not reported.
3. The evaluation is conducted only on a single dataset and limited image resolutions.
4. The visualization results of different methods in Figure5 seems to have little difference.

**Questions:**

1. How does the GNCA strategy scale with larger image sizes and more complex image content?
2. how does GNCA compare to deep learning-based methods in terms of compression performance and computational cost?

---

### Official Review · Reviewer_MzYs · 2024-11-04

**Soundness:** 1
**Presentation:** 3
**Contribution:** 2
**Rating:** 3
**Confidence:** 4

**Summary:**

This paper proposes using Neural Cellular Automata (NCA) models for learned image compression. It proposes a new method Grid Neural Cellular Automata (GNCA), which involves partitioning the images into a grid of patches and training a NCA model on each patch. The model weights serve as the compressed representation of the image. The paper includes results for GNCA models with varying number of parameters and three classic codecs (JPEG, JPEG2000, and WebP) at the highest quality setting on a subset of the COCO dataset.

**Strengths:**

1. The idea of using Neural Cellular Automata (NCA), instead of autoencoders or variational autoencoders (VAEs), for learned image compression is an interesting one and the authors highlight potential advantages of NCA, such as their parallelization capability, model compactness, and robustness to data corruption.
2. The authors propose a novel method for applying NCA to image compression, identifying that images can be spit into grids that are amenable to parallelization.

**Weaknesses:**

1. The experiments are insufficient to compare the proposed NCA method to existing methods. The authors could improve this by:
(a) using more informative metrics (e.g., Rate-distortion curves, which measure the tradeoff between the bits per pixel (bpp) and a distortion metric such as PSNR or MS-SSIM (see Figure 10 in the [1] for an example)). The current paper only compares one compression rate for each of the classic codecs and three compression rates (i.e., model sizes) for the proposed GNCA method.
(b) using datasets that are more common in the image compression community, such as Kodak, Tecnick, or CLIC. The current paper only tests on 30 low resolution images (e.g., 40 x 40 to 120 x 120 pixels) from the COCO dataset.
(c) comparing against other learned image compression models, including [1] and models shown here: https://github.com/InterDigitalInc/CompressAI/tree/master.
2. There is not sufficient evidence for some claims. For example, line 532 states, "The key benefits of the proposed method include its lightweight model architecture, high parallelization capability, and robustness to noise." Since the paper only tests on very small images, it is not clear if these models are more lightweight than other learned image compression models on images of more standard size (e.g., Kodak, CLIC). Similarly, there are no results demonstrating that these models improve robustness to noise. The authors could demonstrate this by comparing NCA models to existing learned image compression models and classic codecs on out-of-distribution or adversarial images.
3. These models are very slow (it takes 3.2 hours to compress a patch of size 40 x 40 pixels). While this is mentioned in the limitations section, this is a serious practical limitation and worth noting.


[1] Wei  Jiang  and  Ronggang  Wang.  Mlic++: Linear  complexity  multi-reference  entropy  modeling for learned image compression.   In ICML 2023 Workshop Neural Compression:  From Information Theory to Applications, 2023.

**Questions:**

1. Can you explain how equation 4 is derived? In particular, what do the 4 and 1 represent?
2. In Figure 4, is the number of parameters for the GNCA models the number of parameters per patch or for the full image? I think line 362 "The total compressed representation size is the sum of the parameters from the NCA models used for each patch" implies that it's the number of parameters for the full image, but I'd like to clarify. Assuming it's for the whole image, can you clarify how the architecture of the patch-NCAs vary as the image size increase?
3. Are there concerns about hallucinations, or outputs that are vastly different (not just lower quality), with NCA models?

---

### Meta-Review · Area_Chair_6Pzt · 2024-12-16

**Metareview:**

The paper proposes a method for image compression using neural cellular automata (NCA). The idea is novel and interesting. However, the evaluation leaves a lot to be desired. Several claims are not well supported by evidence ("low-cost", "lightweight") or not well explained ("robust to noise"). The authors compare only three rate-distortion points for their model with one point for each of their baselines in a table. Not only does this make it difficult to judge the performance of the proposed method, it also suggests a lack of familiarity with the most basic evaluation methods used by the image compression community (e.g., rate-distortion plots).

**Additional Comments On Reviewer Discussion:**

The reviewers pointed out several important shortcomings of the paper. The authors did not submit a rebuttal or interact with the reviewers in any way, hence their criticisms stand.

---

### Decision · Program_Chairs · 2025-01-22

Reject